# Adjustable optical isolator based on the resonant optomechanical interaction

Dong-Yang Wang[*,1] Lei-Lei Yan,[1] Cheng-Hua Bai,[2] Qing He,[1,3] Hong-Fu Wang[†,4] Erjun Liang[‡,1] and Shi-Lei Su[1]

[1]*School of Physics and Microelectronics, Zhengzhou University, Zhengzhou, Henan 450001, People's Republic of China*

[2]*Department of Physics, School of Science, North University of China, Taiyuan, Shanxi 030051, People's Republic of China*

[3]*College of Science, Zhongyuan University of Technology, Zhengzhou, Henan 450001, People's Republic of China*

[4]*Department of Physics, College of Science, Yanbian University, Yanji, Jilin 133002, People's Republic of China*

Optical isolators are indispensable in optical information processing tasks and are essential nonreciprocal devices in chiral networks. We propose a proposal to generate an adjustable bidirectional narrow bandwidth optical isolator based on the optomechanically induced transparency in a spinning whispering gallery microresonator. Analyzing the reason for optical isolation, we find that the nonreciprocity of the system comes from the frequency shift induced by the spinning resonator. To maximize the isolation rate, we devise a way to actively modulate the control field to make the desired optomechanical interaction always resonant. So the realized optical isolator is narrow linewidth, which is evaluated via analytical calculation. Moreover, the location of the optical isolator is related to the angular velocity of the spinning resonator. Our proposal provides a promising strategy for designing adjustable nonreciprocal devices expected to facilitate the study of quantum information processing.

Keywords: optomechanics, optical isolation, optomechanically induced transparency

---

[*] E-mail: dywang@zzu.edu.cn

[†] E-mail: hfwang@ybu.edu.cn

[‡] E-mail: ejliang@zzu.edu.cn

# I.   INTRODUCTION

Cavity optomechanical systems are a powerful platform for studying the interaction between the optical field and the macroscopic mechanical motion [1, 2]. As a bridge connecting the microscopic and macroscopic worlds, the cavity optomechanical system not only plays a vital role in the basic research of quantum physics but also stands out in the field of modern applied science. Many interesting quantum effects have been investigated widely and verified experimentally in the cavity optomechanical system, such as mechanical cooling [3, 4], squeezing [5, 6], entanglement [7, 8], optomechanically induced transparency [9–13], photon blockade [14–16]. Among them, the optomechanically induced transparency is an interesting phenomenon arising from the destructive quantum interference between two different transmission paths in the optomechanical interaction [17–20], which constructs an initially opaque medium and can be used to investigate various effects, such as transmission, fast-slow light, sensor, routing, unresolved sideband cooling, light switch [21, 22].

On the other hand, the optical nonreciprocal effect is an intriguing and important phenomenon in the field of optical research [23], which can be used to develop optical isolators [24], optical circulators [25], directional amplifiers [26], etc. These nonreciprocal optical devices are essential components in quantum networks and quantum information processing, which have attracted much attention in recent years [27]. Such as, the suppression of backscattering has been demonstrated via the nonreciprocity in dielectric resonators [28]. Many researchers have focused on achieving nonreciprocal optical transmission without magnetic materials. So far, various properties to achieve nonreciprocal effects have been proposed, including cavity magnonic interaction [29], thermal motion of atoms [30, 31], Brillouin scattering [32–34], Casimir effect [35], parity-time symmetry [36], effective gauge field [37], spinning microresonator [38, 39]. Usually, these reports can be divided into two different mechanisms to break the Lorentz reciprocity, i.e., phase shift originating from moving medium or reference frame and optical nonlinear effect [23].

Due to superior scalability, the cavity optomechanical system is a good candidate for on-chip nonreciprocity. Some proposals to achieve nonreciprocity have been proposed in various cavity optomechanical systems [40–43]. For example, a reconfigurable nonreciprocal transmission scheme has been reported using purely optomechanical interactions in a superconducting electromechanical circuit [41]. Furthermore, a similar result has been observed

in the silicon optomechanical circuit with optical and mechanical connectivity between two optomechanical cavities [42]. In the previous studies, the nonreciprocal transmission is realized in the loop cavity optomechanical system, composed of more than one set of optical and mechanical modes [44, 45]. In the whispering gallery microresonator, the optomechanically induced nonreciprocal transmission has also been reported via pumping the single direction [46–48]. However, the direction of nonreciprocal transmission strongly depends on the direction of the pumping field. Recently, the effect of a spinning whispering gallery microresonator on the optomechanically induced transparency has been discussed [49], where the researchers found that the optomechanically induced transparency was destroyed and disappeared due to the nonresonant optomechanical interaction. Therefore, it is meaningful to study how to realize a flexible non-directional optical isolator via optomechanically induced transparency.

Here we devise a scheme to realize the flexible non-directional optical isolator based on the optomechanically induced transparency in a spinning whispering gallery microresonator. The frequency of the control field needs to be modulated actively by an acoustic-optic modulator (AOM), which ensures that the beam splitter type optomechanical interaction is always resonant. At this time, the ideal optomechanically induced transparency can be obtained in both directions. The proposal does not require multiple modes to form an optomechanical circuit as in the previous reports [41, 42, 44, 45]. Compared with the schemes that do not consider optomechanical interaction [50] and the optomechanical interaction is nonresonant [49], the linewidth of the optical isolator is much narrower due to the optomechanically induced transparency being alive. Moreover, the direction and location of the optical isolator can also be adjusted by changing the direction of the control field and the angular velocity of the spinning resonator, which is convenient for filtering undesired transmissions and feedback [46, 47].

The rest of the paper is organized as follows: In Sec. II, we illustrate the spinning optomechanical system, derive its Hamiltonian, and analytically calculate the optical transmission in the two opposite directions. In Sec. III, we give the optimal relation to maximize the optical isolation and discuss its linewidth and isolation ratio. Finally, a conclusion is given in Sec. IV.

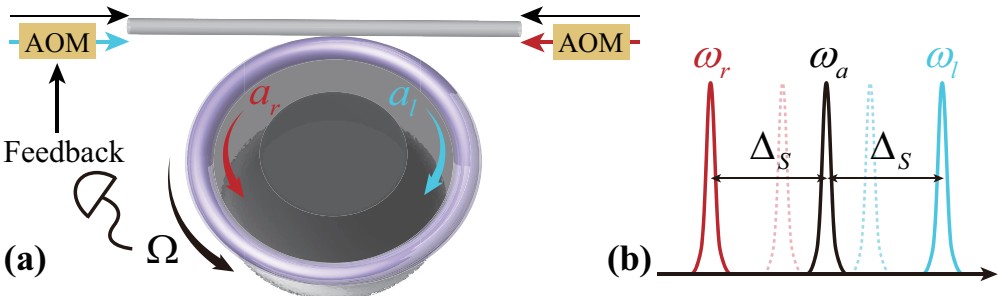

FIG. 1: (a) Schematic diagram of a spinning whispering gallery microresonator coupled to a nearby optical waveguide. The system can be pumped from the two opposite transmission directions by the strong control field (blue and red arrows) and the weak probe field (black arrow), where the control field is modulated by the AOM according to the feedback of angular velocity $\Omega$. (b) The Sagnac-Fizeau frequency shift $\Delta_S$ in the two circulating modes.

## II. MODEL AND HAMILTONIAN

The schematic of our proposal is depicted in Fig. 1(a), where we consider a whispering gallery microresonator coupled to a nearby optical waveguide via the self-adjustment process [50]. The linear optical resonator is mounted on a turbine with the angular velocity $\Omega$, indicating $\Omega$ rotations per second, and the angular velocity $\Omega = 3$ kHz has been reported in experiment [50]. The spinning resonator can induce the Sagnac-Fizeau frequency shift in opposite transmission directions. Thus the resonance frequencies of the two circulating modes are $\omega_{l,r} = \omega_a \pm \Delta_S$ as shown in Fig. 1(b), where $\omega_a$ is the resonance frequency for the stationary resonator and $\Delta_S$ is Sagnac-Fizeau frequency shift induced by the spinning resonator [50, 51]

$$\Delta_S = \frac{nR\Omega\omega_a}{c}\left(1 - \frac{1}{n^2} - \frac{\lambda}{n}\frac{\mathrm{d}n}{\mathrm{d}\lambda}\right), \tag{1}$$

Here, $c$ is the speed of light, $\lambda$ is the wavelength, and $R$ and $n$ are the radius and refractive index of the optical resonance, respectively. The dispersion term is $\mathrm{d}n/\mathrm{d}\lambda \sim 0.01$ in typical materials [52].

First, we consider the case where both the control field and the probe field are input from the left and then discuss the properties of the output field on the right. Considering the mechanical breathing mode induced by the optomechanical interaction, the system

Hamiltonian is written as

$$H = \hbar\omega_l a_l^\dagger a_l + \hbar\omega_r a_r^\dagger a_r + \frac{p^2}{2m} + \frac{1}{2}m\omega_m^2 q^2 - \hbar g\left(a_l^\dagger a_l + a_r^\dagger a_r\right)q$$

$$+i\hbar\varepsilon_c\left(a_l^\dagger e^{-i\omega_c t} - a_l e^{i\omega_c t}\right) + i\hbar\varepsilon_p\left(a_l^\dagger e^{-i\omega_p t} - a_l e^{i\omega_p t}\right), \tag{2}$$

where $p$ and $q$ are the mechanical momentum and displacement operators, respectively. $g$ represents the single-photon optomechanical coupling strength, and the last two terms are the control and probe field interactions, respectively. The control (probe) field amplitude is $\varepsilon_c = \sqrt{2\kappa P/\hbar\omega_c}$ $(\varepsilon_p = \sqrt{2\kappa P_p/\hbar\omega_p})$, where $P$ $(P_p)$ is the pumping power of the control (probe) field, $\kappa$ represents the decay rate of the spinning resonator. The external input fields can be input from both sides, which could determine the direction of optical isolation in our proposal.

By performing a rotating frame defined by $U = \exp[-i\omega_c t a_l^\dagger a_l - i\omega_c t a_r^\dagger a_r]$, i.e., performing the following transformation $a_l(t) \rightarrow a_l(t)e^{-i\omega_c t}$ and $a_r(t) \rightarrow a_r(t)e^{-i\omega_c t}$, the transformed Hamiltonian $H' = U^\dagger HU - i\hbar U^\dagger \dot{U}$ becomes

$$H' = \hbar\Delta_l a_l^\dagger a_l + \hbar\Delta_r a_r^\dagger a_r + \frac{p^2}{2m} + \frac{1}{2}m\omega_m^2 q^2 - \hbar g\left(a_l^\dagger a_l + a_r^\dagger a_r\right)q + i\hbar\varepsilon_c\left(a_l^\dagger - a_l\right)$$

$$+i\hbar\varepsilon_p\left(a_l^\dagger e^{-i\delta_p t} - a_l e^{i\delta_p t}\right), \tag{3}$$

where $\Delta_{l,r} = \omega_a \pm \Delta_S - \omega_c$ and $\delta = \omega_p - \omega_c$ are the detuning of cavity-control field and probe-control field, respectively. Considering the effect of the external thermal environment, the quantum Heisenberg-Langevin equation of the system operator

$$\dot{q} = \frac{p}{m},$$

$$\dot{p} = -m\omega_m^2 q + \hbar g\left(a_l^\dagger a_l + a_r^\dagger a_r\right) - \gamma_m p + \sqrt{2\gamma_m}\xi(t),$$

$$\dot{a}_l = -\left(i\Delta_l + \kappa\right)a_l + iga_l q + \varepsilon_c + \varepsilon_p e^{-i\delta t} + \sqrt{2\kappa}a_l^{\rm in},$$

$$\dot{a}_r = -\left(i\Delta_r + \kappa\right)a_r + iga_r q + \sqrt{2\kappa}a_r^{\rm in}, \tag{4}$$

where $\gamma_m$ is the mechanical damping rate. $\xi(t)$ and $a_{l,r}^{\rm in}$ are the external input noise operators with zero mean value of the mechanical resonator and optical cavity, respectively [53].

Under the mean field approximation, the mean value equations of system operators can be calculated by the factorization assumption, and the results are then written as

$$\langle \dot{q} \rangle = \frac{\langle p \rangle}{m},$$

$$\langle \dot{p} \rangle = -m\omega_m^2 \langle q \rangle + \hbar g \left( \langle a_l^\dagger \rangle \langle a_l \rangle + \langle a_r^\dagger \rangle \langle a_r \rangle \right) - \gamma_m \langle p \rangle,$$

$$\langle \dot{a}_l \rangle = - \left( i\Delta_l + \kappa \right) \langle a_l \rangle + ig\langle a_l \rangle \langle q \rangle + \varepsilon_c + \varepsilon_p e^{-i\delta t},$$

$$\langle \dot{a}_r \rangle = - \left( i\Delta_r + \kappa \right) \langle a_r \rangle + ig\langle a_r \rangle \langle q \rangle. \tag{5}$$

Obviously, the above equation is a set of nonlinear differential equations. Their steady-state solution would contain the Fourier expansion related to the amplitude of the probe field, i.e., $\langle o \rangle = \sum_{k=-\infty}^{+\infty} o_k e^{ik\delta t}$ with $o \in \{q, p, a_l, a_r\}$ and $k \in Z$. Under the condition of a weak probe field, the steady-state response can be truncated to first-order sidebands [9]. At this time, the solution of Eq. (5) is supposed as

$$\langle q \rangle = q_s + q_- e^{-i\delta t} + q_+ e^{i\delta t},$$

$$\langle p \rangle = p_s + p_- e^{-i\delta t} + p_+ e^{i\delta t},$$

$$\langle a_l \rangle = a_{ls} + a_{l-} e^{-i\delta t} + a_{l+} e^{i\delta t},$$

$$\langle a_r \rangle = a_{rs} + a_{r-} e^{-i\delta t} + a_{r+} e^{i\delta t}, \tag{6}$$

Here we have simplified the subscript of the coefficients for convenience, i.e., $o_s = o_0$, $o_- = o_{-1}$, and $o_+ = o_{+1}$. In the original frame without the rotation transformation, $o_s$, $o_-$, $o_+$ correspond to the system responses at frequencies $\omega_c$, $\omega_p$, $2\omega_c - \omega_p$, respectively.

Substituting the above assumption into Eq. (5) and ignoring the higher-order terms, Eq. (5) can be collated into three sets of nonlinear equations, which respectively correspond to the system responses at different frequencies. Under the condition of weak optomechanical coupling and near-resonant red sideband interaction, we can adopt the rotating wave approximation to eliminate the off-resonant blue sideband interaction. Then the cavity

amplitude at the frequency of the probe field is obtained as

$$a_{l-} = \frac{\varepsilon_p \left(\delta^2 - \omega_m^2 + i\delta\gamma_m\right)}{\left(\delta^2 - \omega_m^2 + i\delta\gamma_m\right)\left[\kappa - i\left(\delta - \Delta\right)\right] + 2i\omega_m\beta}, \tag{7}$$

where $\Delta = \Delta_l - 2\beta/\omega_m$, $a_{ls} = \varepsilon_c/(i\Delta + \kappa)$, $\beta = \hbar g^2|a_{1s}|^2/(2m\omega_m)$. In the above calculation, the non-resonant interaction in the system has been ignored by the usual rotating wave approximation method. Therefore, the obtained result is different from the previous works, which would cause to the disappearance of the slightly optomechanically induced amplification [9, 11, 12]. Next, the output field can be derived easily by utilizing the usual input-output relationship, i.e., $a_{\text{out}} = a_{\text{in}} - \sqrt{\kappa}a_{l-}$, and then the transmission rate of the probe field is given by $T_l = |a_{\text{out}}/a_{\text{in}}|^2$. Because the system works in the near-resonant region of the red sideband, i.e., $\Delta \sim \delta \sim \omega_m$, the transmission rate of the probe field located at the near-resonant region can be simplified to

$$T_l = \frac{4\left[\beta - \left(\delta - \omega_m\right)^2\right]^2 + \gamma_m^2\left(\delta - \omega_m\right)^2}{4\left(\delta - \omega_m\right)^4 + \left(4\kappa^2 - 8\beta + \gamma_m^2\right)\left(\delta - \omega_m\right)^2 + \left(2\beta + \gamma_m\kappa\right)^2}. \tag{8}$$

Of course, the optomechanically induced transparency is existence when the system works in the near-resonant region of the blue sideband, where the destructive quantum interference can also occur. At this time, the final transmission rate is slightly different from that in the red sideband case. Moreover, when the probe field is input from the right, the transmission rate $T_r$ calculation is similar and will not be repeated here.

## III.  OPTICAL ISOLATOR

### A.  Nonreciprocity

Based on the above results about the transmission rate on both sides, we first discuss the nonreciprocity of the system. Before that, we need to study the principle of nonreciprocity and discuss how to maximize the optical isolation ratio in our proposal.

First, it is essential to emphasize that the frequency of control field $\omega_c$ is dependent on the rotation velocity of the resonator in our proposal so that the red sideband of optomechanical interaction is always resonant, i.e., $\omega_l - \omega_c \sim \omega_m$, which ensures that the optomechanically induced transparency is always ideal even in the case of different angular velocities. The relationship between the pumping frequency of the control field and the angular velocity is

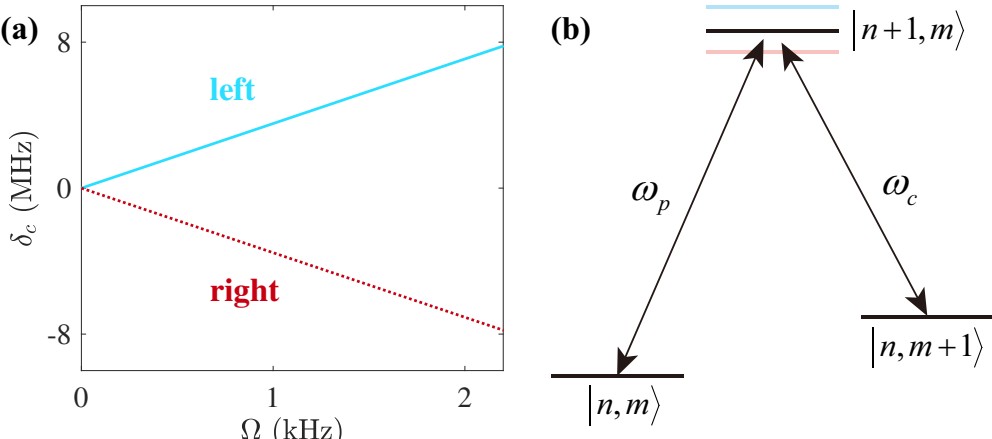

FIG. 2: (a) The pumping frequency shift $\delta_c$ of the control field in two opposite directions versus angular velocity. Here, according to the relevant experiments [50, 52, 54, 55], we set the system parameters $n = 1.48$, $R = 1.1$ mm, $m = 10$ ng, $\lambda = 1550$ nm, $Q = 6 \times 10^8$, $\kappa = 20$ MHz, $\omega_m = 63$ MHz, $\gamma_m = 10$ kHz, $T = 130$ mK, $P = 5$ mW. (b) Schematic of the energy-level diagram, where $|n, m\rangle$ denotes the system's state with $n$ photons and $m$ phonons.

shown in Fig. 2(a). For the input control field pumping from the left, its frequency needs to increase with the rotation velocity, as the blue solid line shown in Fig. 2(a). However, when the control field is input from the right, we have to appropriately decrease the frequency of the control field to make the red sideband resonant, as shown by the red dotted line. Thus, the control field frequency needs to be changed in the range of a few megahertz $(0 \sim 8$ MHz), which can be achieved by the frequency shift devices. The system parameters in our proposal are chosen according to the relevant experiments [50, 52, 54, 55], which can keep the system from unstable and multi-stable regions. On the other hand, different from the standard calculation about optomechanically induced transparency [9, 56, 57], the above result has utilized the rotation wave approximation to ignore the effect of off-resonant blue sideband interaction under the condition of weak coupling. Therefore, the model is finally reduced to a three-level system, as shown in Fig. 2(b), whose mechanism is similar to electromagnetic induced transparency. So it also results in the disappearance of the optomechanically induced amplification phenomenon, which is extremely tiny in the weak coupling regime.

To intuitively show the phenomenon of nonreciprocal optical transmission, as shown in

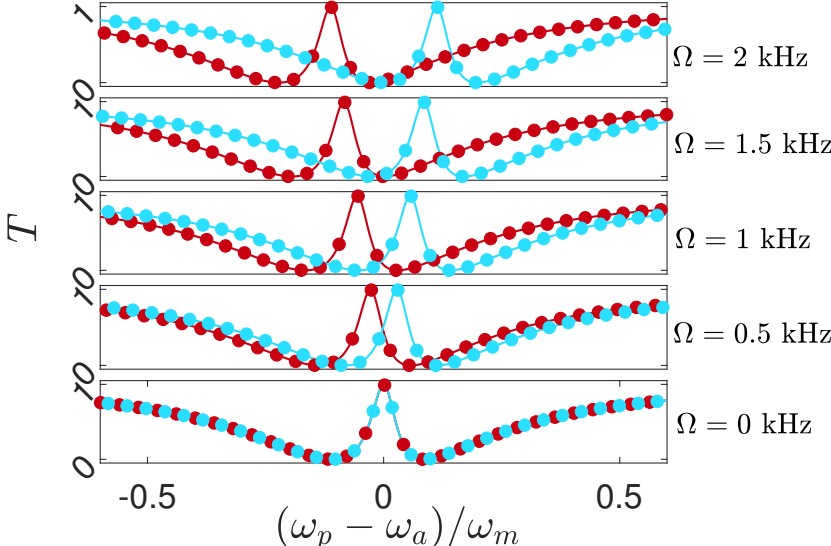

FIG. 3: The transmission rate of two opposite directions versus the probe field detuning with different angular velocities. Here, the blue (red) lines represent the transmission rate when the probe field is input from the left (right). And the angular velocity is set respectively as $\Omega = \{0 \text{ kHz}, 0.5 \text{ kHz}, 1 \text{ kHz}, 1.5 \text{ kHz}, 2 \text{ kHz}\}$ from bottom to top, which is feasible for the current experiment [50]. And the system parameters are chosen the same as those in Fig. 2.

Fig. 3, we plot the transmission rate of the two opposite transmission directions with different angular velocities. Since the frequency shift caused by the rotation of the spinning resonator increases with the angular velocity increasing as in Eq. (1), the window of optomechanically induced transparency gradually separates to both sides. Here, it is worth noting that the ideal optomechanically induced transparency does not disappear even in either direction with the appearance of cavity frequency shift. That is mainly because the control field frequency varies with the angular velocity, which ensures the red sideband of optomechanical interaction is always resonant. The optomechanically induced transparency is preserved by actively modulating the control field frequency. Otherwise, the optomechanically induced transparency would be destroyed and disappear as previous reports [49, 51]. The frequency shift of the control field can be achieved by the normal AOM in experiments [58, 59]. In Fig. 3, we can also find an optimal value for the angular velocity of the spinning resonator in the case of taking into account optical isolation and transmission rate, which would maximize the difference in transmission rates between the two directions. Based on those selected system parameters, the optimal value of the angular velocity is $\Omega \simeq 800$ Hz. When

the external control field is weak, the relationship between the optimal angular velocity and the system parameters can be given by

$$2\Delta_S = \frac{1}{2}\sqrt{\frac{2\sqrt{\beta\left[4\beta\left(\gamma_m+\kappa\right)^2+\gamma_m^2\kappa\left(\gamma_m+2\kappa\right)\right]}-\gamma_m\left(4\beta+\gamma_m\kappa\right)}{\kappa}}, \tag{9}$$

where the right part is the distance between two adjacent real roots of $\partial T_l/\partial\delta = 0$. This indicates that the optimal angular velocity also needs to be changed accordingly for different system parameters.

## B. Linewidth of optical isolator

In the spinning resonator without optomechanical interaction, the effect of irreversible refraction is achieved due to the Sagnac-Fizeau frequency shift [50]. The linewidth of optical isolation is twice the cavity decay rate. Here, we will use the optomechanically induced transparency to significantly reduce the linewidth of the optical isolator, which is helpful for transmitting signals of a specific frequency.

In our proposal, since the nonreciprocal transmission comes from the optomechanical interactions [10], the linewidth of the optical isolator is much narrower than that in the normal spinning resonator system without mechanical motion [50], as shown in Fig. 4(a). When the control field is absent ($P = 0$), the input probe field is resonantly absorbed in the frequency of clockwise cavity $\omega_p \sim \omega_a + \Delta_S$. Meanwhile, the nonreciprocal transmission still occurs only when the Sagnac-Fizeau frequency shift caused by the spinning resonator is larger than the cavity decay rate. Moreover, the linewidth of optical isolation is $2\kappa$, which is only related to the cavity decay rate [50]. However, when the system is driven by the red detuning control field, the resonant optomechanical interaction will induce a narrow linewidth transparent window. At this time, the optical transmission is reversed due to the optomechanically induced transparency, which helps filter undesired transmissions and feedback. Then, the linewidth (full width at half maximum) of the optical isolator is given by

$$\Gamma = \gamma_m + \frac{2\beta}{\kappa}, \tag{10}$$

which can be modulated by the pumping power of the control field, as shown in Fig. 4(b).

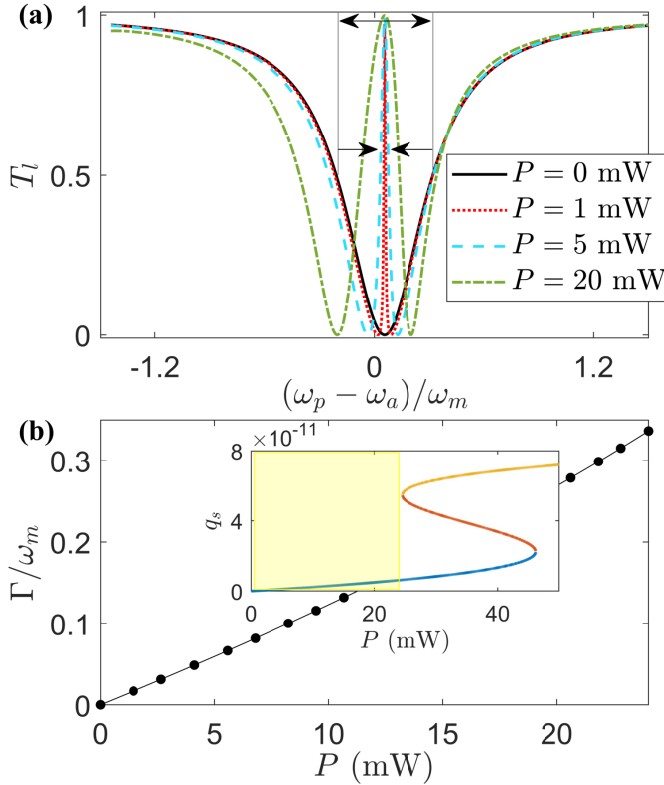

FIG. 4: (a) The transmission rate of left input versus the probe field detuning with different pumping powers. The optical transmission is reversed when the pumping field drives the spinning resonator. (b) The linewidth of the optical isolator versus the pumping power of the control field. Here, the inset represents the relationship between the steady-state mechanical displacement and the pumping power, which indicates that the proposal of the optical isolator is always in the monostable region. The other parameters are chosen the same as those in Fig. 2.

It is worth noting that the pumping power does not increase arbitrarily, which is always controlled to be less than the bistable boundary $P < 25$ mW in our proposal. At this time, the optical transmission spectrum is approximately symmetrical, and the linewidth is narrow enough. To demonstrate the stability of our proposal, we plot the mechanical displacement as a function of driving power in the inset of Fig. 4(b).

### C. Isolation ratio

Next, to evaluate the efficiency of this proposal, it is convenient to redefine an optical isolation ratio

$$\eta = 10 \log \left( \frac{T_l}{T_r} \right), \tag{11}$$

where $T_l$ and $T_r$ are the transmission rates input from the left and right sides, respectively. As shown in Fig. 5, the center planform shows the change of the redefined isolation ratio $\eta$ with the probe field detuning $(\omega_p - \omega_a)/\omega_m$ and the angular velocity $\Omega$. Here, the two different colors correspond to the isolation ratio in the two opposite transmission directions, which can certify that the non-directional optical isolator is achieved.

Moreover, the location of isolation also changes with the angular velocity. We can see that the higher gray-scale area corresponds to the higher isolation ratio. However, it is worth pointing out that the high isolation ratio does not mean a significant difference between the transmission rates of both sides. The high isolation ratio comes from the minimal transmission rate on one side, almost equal to 0, as seen from the surrounding insets in Fig. 5. The ideal optical isolator, possessing both the high isolation rate and the transmission rate close to 1 on one side, only occurs in the 'Λ'-type part in the middle of Fig. 5. Meanwhile, the transmission rates on both sides are shown as the two insets at the bottom. The above results show that the non-directional optical isolator is achieved, and the location of optical isolation is adjustable by changing the angular velocity of the spinning resonator, which is helpful for realizing directed optical information processing and building chiral networks.

Finally, we briefly discuss the experimental feasibility of the spinning microresonator system. The proposed proposal is demonstrated in a whispering gallery microresonator, which can be mounted on a turbine. The used angular velocity in our proposal $\Omega = 3$ kHz has been realized in the recent experiment [50]. The optomechanically induced transparency has also been observed experimentally [10, 60, 61]. The needed frequency shift of the control field is small and can be achieved by AOM in experiments [58, 59]. The system parameters are chosen reasonably according to the relevant experiments, and the system is always in the monostable region.

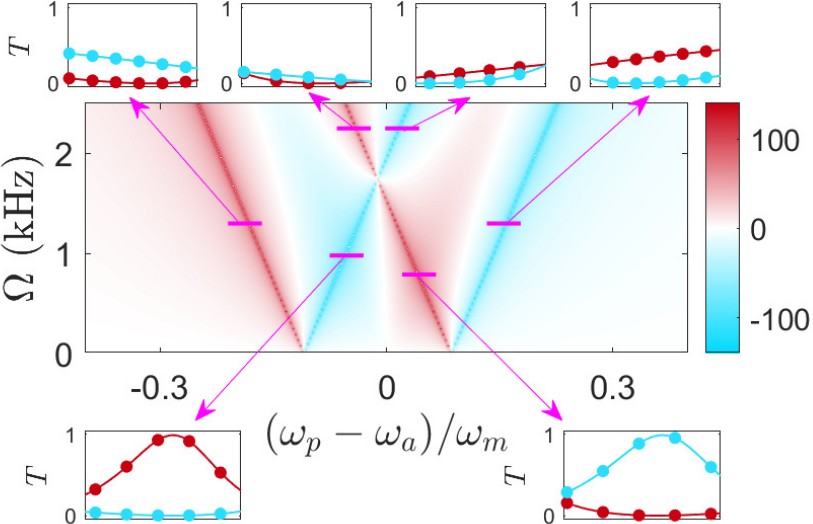

FIG. 5: The optical isolation ratio $\eta$ versus the probe field detuning and the angular velocity. The surrounding insets represent the transmission rate of two opposite directions at corresponding positions. The other parameters are chosen the same as those in Fig. 2.

## IV. CONCLUSIONS

In conclusion, we have investigated how to achieve an adjustable non-directional optical isolator in the spinning optomechanical resonator. In our proposal, the optomechanically induced transparency is preserved by actively modulating the control field frequency so that the desired optomechanical interaction is always resonant and is no longer affected by the optical frequency shift. Based on experimentally feasible devices and parameters, we obtain the optical isolator with high isolation and narrow linewidth, which helps filter unwanted transmissions and feedback in various optical transmission tasks. Moreover, the location of transparent windows can also be adjusted by changing the angular velocity of the spinning resonator and the pumping power of the control field. Our proposal provides flexible options for an ideal non-directional optical isolator, which is helpful for different optical transmission tasks.

### Acknowledgements

This work was supported by the National Natural Science Foundation of China under Grants (No. 12147149, No. 12204424, No. 12274376, No. 12074346, No. U21A20434),

the China Postdoctoral Science Foundation (No. 2022M722889), and the Natural Science Foundation of Henan Province under Grant (No. 212300410085).

---

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
