# Peer review of "Adjustable optical isolator based on the resonant optomechanical interaction"

_SciPost Physics_

## Round 1 · Referee Report · Anonymous (Referee 1) · 2023-2-10

Report

The authors propose intriguing phenomena to observe nonreciprocal behaviors in optomechanical resonators by rotating systems. Specifically, the Sagnac effect induced by rotation enables nonreciprocal shifts of optomechanically induced transparency windows, differentiating the response of S12 from that of S21. Although the quality of presented figures is in general descent, the text must be improved language-wise. The authors must formally rewrite the texts and minimize grammatical errors, as I have encountered too many language errors when reading the manuscript. Also, I have some suggestions and comments, so I would like the authors to address them before recommending the manuscript be published in SciPost Physics.
1) Equation (10) is incorrect since it is only valid for the red-detuned case. The authors should correctly define q_s again in Eq. (10).
2) The authors claimed that the optimal angular velocity is around 800 Hz for optical isolation, and this value depends on the parameters. Is there any way to normalize the value by certain parameters?
3) Missing unit for \Omega in Figure 3.
4) The authors mentioned that '… where we consider a whispering gallery microresonator coupled to a nearby optical waveguide via the self-adjustment process [36].' I believe Tal Carmon's Nature [47] paper thoroughly studied how a tapered fiber could be stably coupled to a spherical resonator, not this theory paper [36].
5) The authors can cite another paper experimentally showing optical isolations by Brillouin optomechanics followed by [31]
J. Kim, S. Kim, and G. Bahl, "Complete linear optical isolation at the microscale with ultralow loss," Scientific reports 7 (1), 1647, 2017.
And can cite the interesting paper demonstrating the suppression of backscattering through the nonreciprocal response by the optomechanical interactions.
S. Kim, J.M. Taylor, and G. Bahl, "Dynamic suppression of Rayleigh light scattering in dielectric resonators," Optica 6 (8), 1016-1022, 2019.

  • validity: -
  • significance: -
  • originality: -
  • clarity: -
  • formatting: -
  • grammar: -

Author:  Dong-Yang Wang  on 2023-02-23  [id 3396]

(in reply to Report 1 on 2023-02-10)

Q1) Equation (10) is incorrect since it is only valid for the red-detuned case. The authors should correctly define q_s again in Eq. (10).
Reply) Many thanks for your reminder. We have corrected Eq. (10) and added the corresponding descriptions in our revised manuscript. It should be noted that most of the results in our manuscript are discussed in the red-detuned case.
Q2) The authors claimed that the optimal angular velocity is around 800 Hz for optical isolation, and this value depends on the parameters. Is there any way to normalize the value by certain parameters?
Reply) Thanks for your suggestion. We have given a relationship between the optimal angular velocity and the system parameters in Eq. (9) of our revised manuscript. To give a simpler relationship, we recalculate the transmission rate when the system works in the near-resonant region of the red sideband, and please see Eq. (8) in the revised manuscript. So the relationship is only valid for the weak control field case.
Q3) Missing unit for \Omega in Figure 3.
Reply) Thanks for your reminder. We have revised it in our manuscript.
Q4) The authors mentioned that '… where we consider a whispering gallery microresonator coupled to a nearby optical waveguide via the self-adjustment process [36].' I believe Tal Carmon's Nature [47] paper thoroughly studied how a tapered fiber could be stably coupled to a spherical resonator, not this theory paper [36].
Reply) Thanks for your reminder. We have revised the reference in our manuscript.
Q5) The authors can cite another paper experimentally showing optical isolations by Brillouin optomechanics followed by [31] J. Kim, S. Kim, and G. Bahl, "Complete linear optical isolation at the microscale with ultralow loss," Scientific reports 7 (1), 1647, 2017. And can cite the interesting paper demonstrating the suppression of backscattering through the nonreciprocal response by the optomechanical interactions. S. Kim, J.M. Taylor, and G. Bahl, "Dynamic suppression of Rayleigh light scattering in dielectric resonators," Optica 6 (8), 1016-1022, 2019.
Reply) Thanks for your suggestion. We have cited the references in the introduction part of our revised manuscript.
We have carefully checked and revised the English language of our manuscript. Thanks again for your valuable suggestions and comments, which are pretty instructive for our manuscript!

---

## Round 2 · Referee Report · Anonymous (Referee 1) · 2023-5-17

Report

The authors have revised their manuscript to address the concerns and issues raised by the referee. The clarity of the discussion is certainly improved. In my view, the revised manuscript is ready for publication in SciPost.

---

## Round 2 · Referee Report · Anonymous (Referee 2) · 2023-7-18

Report

In their submission the authors address the question how to realise a "non-directional optical isolator" via optomechanical induced transparency. They propose a pump scheme for an optomechanical system consisting of two cavities and one mechanical oscillator and use standard tools to calculate the transmission amplitudes and isolation ratios.

First of all, the revised submission still contains language oddities, e.g. to propose a proposal. Importantly, I wonder if the authors really mean "non-directional optical isolator" and not "directional" or "non-reciprocal" which would make more sense to me.

I found the literature review to be partially inaccurate in important places, e.g. Ref. 40-42 are cited for "proposals" whereas all three papers actually reported experiments!

The theoretical analysis given in this submission (Eq. 3 to 6) is absolutely standard. The only result that is new, at least in the current form, is Equation (7), and closely related Equation (8). These quantities are then shown/discussed in Figure 3 through 5.

The reader is told that the narrow bandwidth of the proposed isolator will be an asset but of course signals have finite bandwidth and a very narrow filter is not useful in practice. The device has to fit its purpose.

In my opinion the reported results, though they are likely technically correct, are not significant enough to warrant publication in any of the SciPost journals.
  • validity: ok
  • significance: low
  • originality: poor
  • clarity: ok
  • formatting: acceptable
  • grammar: below threshold

---

## Round 2 · Author Response

The optical isolator allows the transmission of light in only one direction and prevents unwanted feedback, which plays an important role in optical information processing tasks. Here we propose a scheme to generate an adjustable ultra-narrow optical isolator based on the optomechanically induced transparency in a spinning whispering gallery microresonator. Compared with previous works, our designed scheme has the following characteristics:
(1) The direction and location of the optical isolator can be adjusted in our scheme.
(2) The linewidth of the optical isolator is narrow, which helps filter unwanted transmissions and feedback in various optical transmission tasks.
(3) The optomechanically induced transparency is preserved by actively modulating the frequency of the control field.
(4) The realized adjusted non-directional optical isolator can provide flexible options for different optical transmission tasks.

---

## Round 2 · List of Changes

The detailed responses are attached as follows, and the main changes are also marked in blue in our revised manuscript.
Q1) Equation (10) is incorrect since it is only valid for the red-detuned case. The authors should correctly define q_s again in Eq. (10).
Reply) Many thanks for your reminder. We have corrected Eq. (10) and added the corresponding descriptions in our revised manuscript. It should be noted that most of the results in our manuscript are discussed in the red-detuned case.
Q2) The authors claimed that the optimal angular velocity is around 800 Hz for optical isolation, and this value depends on the parameters. Is there any way to normalize the value by certain parameters?
Reply) Thanks for your suggestion. We have given a relationship between the optimal angular velocity and the system parameters in Eq. (9) of our revised manuscript. To give a simpler relationship, we recalculate the transmission rate when the system works in the near-resonant region of the red sideband, and please see Eq. (8) in the revised manuscript. So the relationship is only valid for the weak control field case.
Q3) Missing unit for \Omega in Figure 3.
Reply) Thanks for your reminder. We have revised it in our manuscript.
Q4) The authors mentioned that '… where we consider a whispering gallery microresonator coupled to a nearby optical waveguide via the self-adjustment process [36].' I believe Tal Carmon's Nature [47] paper thoroughly studied how a tapered fiber could be stably coupled to a spherical resonator, not this theory paper [36].
Reply) Thanks for your reminder. We have revised the reference in our manuscript.
Q5) The authors can cite another paper experimentally showing optical isolations by Brillouin optomechanics followed by [31] J. Kim, S. Kim, and G. Bahl, "Complete linear optical isolation at the microscale with ultralow loss," Scientific reports 7 (1), 1647, 2017. And can cite the interesting paper demonstrating the suppression of backscattering through the nonreciprocal response by the optomechanical interactions. S. Kim, J.M. Taylor, and G. Bahl, "Dynamic suppression of Rayleigh light scattering in dielectric resonators," Optica 6 (8), 1016-1022, 2019.
Reply) Thanks for your suggestion. We have cited the references in the introduction part of our revised manuscript.
We have carefully checked and revised the English language of our manuscript. Thanks again for your valuable suggestions and comments, which are pretty instructive for our manuscript!

---

## Editorial Decision

awaiting_resubmission